# Chemosensory continuity from prenatal to postnatal life in humans: A systematic review and meta-analysis

**Beyza Ustun** *, **Judith Covey, Nadja Reissland**

Department of Psychology, Durham University, Durham, United Kingdom

* beyza.n.ustun@durham.ac.uk

## Abstract

Throughout pregnancy, fetuses are exposed to a range of chemosensory inputs influencing their postnatal behaviors. Such prenatal exposure provides the fetus with continuous sensory information to adapt to the environment they face once born. This study aimed to assess the chemosensory continuity through a systematic review and meta-analysis of existing evidence on chemosensory continuity from prenatal to first postnatal year. Web of Science Core. Collections, MEDLINE, PsycINFO, EBSCOhost ebook collection was searched from 1900 to 2021. Studies identified from the search were grouped according to type of stimuli the fetuses were exposed to prenatally that the neonatal infants' responses to were being evaluated, namely flavors transferred from the maternal diet, and the odor of their own amniotic fluid. Of the 12 studies that met the eligibility criteria for inclusion ($k = 6$, $k = 6$, respectively in the first and the second group of studies), and eight studies ($k = 4$, $k = 4$, respectively) provided sufficient data suitable for meta-analysis. Infants, during their first year of life, oriented their heads for significantly longer durations in the direction of the prenatally experienced stimuli with large pooled effect sizes (flavor stimuli, $d = 1.24$, 95% CI [0.56, 1.91]; amniotic fluid odor, $d = 0.853$; 95% CI [.632, 1.073]). The pooled effect size for the duration of mouthing behavior was significant in response to prenatal flavor exposure through maternal diet ($d = 0.72$; 95% CI [0.306, 1.136]), but not for the frequency of negative facial expressions ($d = -0.87$, 95% CI [-2.39, 0.66]). Postnatal evidence suggests that there is a chemosensory continuity from fetal to the first year of postnatal life.

## Introduction

Fetuses are reactive to their environment in the womb based on their developing sensory abilities [1, 2], which allow postnatal detection of, for example, the odor of amniotic fluid, and flavor cues from the diet of the pregnant mother [3]. Research suggests that fetal experiences have an impact on the behavior of infants [4–6]. Therefore, it is argued that postnatal reactions are influenced after birth not only by perceptual capacities, but also by prenatal sensory inputs and the ability to embed, and access this learning when faced with stimuli familiarized in the womb [1, 7].

**Data Availability Statement:** All relevant data are within the paper and its Supporting Information files.

**Funding:** This study was undertaken as part of a Ph.D. thesis funded by the Turkish Ministry of

National Education. The funder has not had any role in the study protocol, analysis, or preparation of the manuscript.

**Competing interests:** The authors have declared that no competing interests exist.

Transnatal chemosensory transmission, also known as transnatal chemosensory continuity, refers to the transition from intra-uterine to extra-uterine life [1, 7]. This continuity, because of early familiarization, facilitates new-born infants' ability to adapt to their postnatal environment [3, 8, 9]. Taste, olfaction, and trigeminal chemesthesis cannot be dissociated in the intra-uterine environment, and thus we refer to flavor exposure acknowledging that their effects on the fetus may involve one or several of these chemosensory inputs [10]. Postpartum reactions, that provide indirect evidence of chemosensory transmission, to such prenatal flavor exposure, may be observed for weeks, months, and potentially years after birth.

It is essential to understand how the evidence supporting transnatal continuity is to be assessed and managed in practice. Such a process might facilitate programming healthy behaviors since sensory abilities are already functional at the fetal stage [1, 11]. Different postnatal behaviors, food acceptance or preference for familiar flavor, can be attributed to prenatal perception of the formation of discriminative abilities [12].

Existing relevant systematic reviews [13, 14] have been conducted assessing the effects of pre and/or postnatal flavor exposure on postnatal behavioral outcomes until the age of 2 years old [14] and the age of 9 years old [13]. Their results showed that infants can recall the volatiles transmitted to amniotic fluid from the maternal diet. In our systematic review, in addition to the infant reactions to the flavors transmitted via maternal diet, we also investigated infant responses to the odor of their own amniotic fluid which would provide further evidence of chemosensory continuity from prenatal to postnatal life. The current review is the first undertaking a metaanalysis to determine pooled summary effect sizes of the chemosensory transmission from fetal life to the first postnatal year of life, in human participants. Unlike previous reviews, we focused on the first postnatal year since experiences during this critical stage of development can have long-term consequences, especially in terms of food-related behaviors [15–17].

Our systematic review and meta-analysis was therefore designed to synthesize the findings from two related bodies of research: 1) studies that have investigated whether infants (Participant) have different behavioral profiles (Outcome) when they are exposed to flavor stimuli (Exposure) via maternal diet during pregnancy, and 2) studies that have investigated whether infants (Participants) have different behavioral profiles (Outcome) to the odor of their amniotic fluid that they have been exposed to through gestation (Exposure). In the first group of studies, we compared the behavioral responses of infants whose mothers ingested the target flavor to those who did not experience the same flavor. In the second group of studies, infant reactions to the odor of familiar amniotic fluid odor (collected from their own mother) were compared either to their reactions to the odor of unfamiliar amniotic fluid (collected from a different mother) odor or to a control odor such as distilled water.

## Methodology

The methods used for our systematic review and meta-analysis followed the PRISMA guidelines [18].

### Search methods for identification of the studies

A literature search was conducted using four electronic databases (Web of Science Core Collections, MEDLINE, PsychINFO and EBSCOhost ebook collection) for studies conducted in the date range 1900 and December 2021. The search terms are presented in Table 1. The reference list of all papers identified from the keyword search, were manually screened to identify any further studies of interest to ensure literature saturation.

**Table 1. Search terms.**

| Prenatal terms | AND | Chemical sense and food terms | AND | Exposure or continuity terms | AND | Subject terms | AND | Behavior terms | NOT | Animal terms |
|---|---|---|---|---|---|---|---|---|---|---|
| Prenatal | | Taste* | | Exposure | | F$etus* | | Preference* | | Animal* |
| OR | | OR | | OR | | OR | | OR | | OR |
| Early | | flavo$r* | | Experience | | Baby* | | Acceptance | | Cat* |
| OR | | OR | | OR | | OR | | OR | | OR |
| Pregnancy | | odo$r | | Learning | | Infant* | | behavi$r* | | Dog* |
| OR | | OR | | OR | | OR | | OR | | OR |
| Intrauterine | | Chemical sense* | | Perception | | Newborn* | | facial expression* | | Bird* |
| OR | | OR | | OR | | OR | | OR | | OR |
| in the womb | | Smell | | Ingestion | | Postnatal | | Response* | | Mice* |
| OR | | OR | | OR | | OR | | OR | | OR |
| Perinatal | | Gustatory | | Transnatal continuity | | Neonate* | | facial movement* | | Rat* |
| OR | | OR | | OR | | | | OR | | OR |
| Maternal | | Olfactory | | Transnatal transmission | | | | Liking | | Piglet* |
| OR | | OR | | | | | | OR | | OR |
| Pregnant woman* | | Amniotic fluid | | | | | | Disliking | | Mammal* |
| OR | | OR | | | | | | OR | | OR |
| Mother* | | Food | | | | | | Intake | | Monkey* |
| | | OR | | | | | | OR | | OR |
| | | beverage* | | | | | | Refusal | | Lamb* |
| | | OR | | | | | | OR | | OR |
| | | dietary supplement* | | | | | | Appetite OR Attraction | | Rabbit* |

*Note.* "AND" showing that studies required having one term from each column, "OR" showing that any of those terms is adequate for eligibility, "NOT" showing that any of those terms is adequate for ineligibility.

## Selection of studies

The database searches were imported into EndNote X20. Duplicates were automatically removed while importing. Before the screening, one author (BU) checked and removed any remaining duplicates manually. To ensure that there was no double counting, we coded the studies with the same cohort of participants published in multiple publications as a single study. The primary author (BU) examined the titles and abstracts of studies to evaluate their fit to the inclusion criteria. Full texts of relevant studies were screened for further analysis of inclusion criteria, with a second reviewer (JC or NR), consulted to resolve any discrepancies.

## Inclusion and exclusion criteria

The target population was healthy (as indicated in the studies) mother-infant dyads from prenatal life to the first postnatal year.

In the first group of studies that reviewed the effects of specific flavor exposure studies were included if they reported a measure of a specific prenatal flavor exposure transferred via maternal diet and postnatal behavioral responses (i.e., orofacial reactions and head orientation) to that flavor. Infant head orientation to a stimulus experienced and rehearsed in the prenatal environment indicates a preference for the stimulus [19–21]. Furthermore, orofacial responses have been found to be a robust indication of hedonic discrimination of infants up to one-year-old [19–21]. Studies were included if they reported a comparison between groups of

infants ≤ 1-year if age with and without prenatal exposure to the specific flavor exposure. Inclusion criteria for studies were not restricted to randomized designs. No restrictions on inclusion were applied for duration or type of flavor exposure during pregnancy, fetal age at time of exposure, type of postnatal stimuli (taste, odor, or both), or type of maternal intake route (e.g., intake via food, beverages, or dietary supplement). Since breastmilk contains maternal dietary aromas [22], the results from the breastfeeding period were excluded if a study evaluated the effects of a particular flavor exposure during both pregnancy and breastfeeding.

In the second group of studies on the infants 'responses to the odor of their own amniotic fluid, studies were included if they reported a measure of infant behaviors (i.e., orofacial responses and head orientation) at ≤ 1-year after birth in response to the odor of their own amniotic fluid compared to their response to a control condition such as distilled water or amniotic fluid from another mother. Inclusion criteria for studies were not restricted to randomized designs. If different comparators were used in a study and one of the comparators involved pairing with distilled water, this control odor was chosen in the analysis because this type of water is purified and devoid of contaminants.

Both groups of studies in the systematic review excluded: animal studies where animals were a total or a part of the sample of a study, studies that were not published in the English language, unpublished studies, reviews, meta-analyses, letters, opinions, conference or poster abstracts, studies focusing on medical/ health or birth outcomes, studies reporting unhealthy samples with diagnosed disease or condition. Studies reporting mothers with gestational diabetes, mothers with allergies, obesity, or hyperemesis were excluded as were studies reporting fetal anomalies at 12 or 20 weeks, malnourished fetuses, preterm delivery (< 37 weeks), or low birth weight (≤ 2500g). Studies involving pregnant mothers who knowingly smoked or used Nicotine Replacement Therapy were also excluded. In longitudinal studies where infant behaviors were assessed at multiple points, we selected the earliest time point to capture the earliest infant reactions after birth. No restrictions were applied regarding maternal age, race, ethnicity, socioeconomic status, parity, or study sample size.

## Data extraction and management

All data were extracted in a pre-defined form by the primary author (BU) and were confirmed by the other authors (JC and NR). The data extraction form summarized study and participant characteristics (sample size, infant age at testing, when applicable gestational age during exposure), infant behavioral measurement, outcomes, study effect sizes and controlled confounders. The form also extracted information about the flavor type, testing stimulus, and type of control condition, if relevant.

## Assessment of risk of bias (RoB) in included studies

The risk of bias of each included study was assessed using either ROBINS-I or ROB 2 tools [23, 24] which are recommended by the Cochrane Collaboration. All studies were checked for overall risk-of-bias judgement with a range of components (See S1 Table). Each component was rated as low risk, some concern, and high risk. Any disagreements were discussed in consultation with the other authors (JC or NR) until all disagreements were resolved.

## Data synthesis and analysis

First, we provide a narrative descriptive summary of the findings from the studies included in this systematic review. This narrative overview summarizes the main findings across studies based on postnatal behavioral outcomes of infants when re-exposed to prenatal stimuli.

Studies were categorized with respect to infant responses to prenatal chemosensory stimuli. If there was a minimum of two studies [25] using the same infant assessment method and the studies provided sufficient data to calculate effect sizes, a meta-analysis was carried out using Comprehensive Meta-Analysis [26]. Effect sizes (Cohen's d values) were calculated for each study and Cohen's convention was used to assess effect sizes, with 0.2 indicating a small effect, 0.5 a moderate effect, and 0.8 indicating a large effect [27]. Heterogeneity was assessed using Cochran's Q and $I^2$. We concluded there was evidence for heterogeneity when the p-value for Cochran's Q was significant ($p < .05$) and if the $I^2$ was greater than 50% [28]. A fixed effect or random effects model was reported depending on whether or not the effect sizes were homogeneous. The number of missing studies that would need to be retrieved for the effect size to be non-significant was estimated using Rosenthal's Fail-safe N [29].

## Results

### Study selection

The literature search yielded 1,927 potentially relevant articles. After removing duplicates (automatically and manually) and ineligible records depending on language and publication type 1,673 studies remained. Having examined the title and abstract of those studies, 57 full-text articles were evaluated in detail. Of those, 45 were excluded, leaving a total of six studies [30–35] meeting the eligibility criteria for the first group of studies and six studies [36–41] meeting the inclusion criteria for the second group of studies. The most common reason for exclusion from the systematic review was using postnatal exposure to flavor as an independent variable ($k = 18$). Because of sufficient results reported and having more than one study to conduct a meta-analysis, four studies [32–35] were included in the first group (flavor stimuli from maternal diet) and four studies [36–39] were included in the second group (amniotic fluid odor). Where possible, the authors of the studies reporting insufficient data were contacted to obtain further information. Fig 1 shows the PRISMA flowchart of the search strategy and study selection. The overall risk of bias in included studies was mostly judged to be low, only one study [32] was judged to have some concerns due to not reporting confounding variables (S1 Table).

### Study characteristics

Studies included in the review came from different countries: France ($k = 8$), Argentina ($k = 2$), USA ($k = 1$), and Northern Ireland ($k = 1$). There were two randomized controlled studies [33, 41] and one non-RCT (within-between subject) [34], the remainder were longitudinal cohort studies [30–32, 35–40]. Characteristics of these studies are described in Tables 2 and 3.

### Association between prenatal flavor exposure through maternal diet and infant behaviors

Table 2 shows the six studies [30–35] that had investigated whether infants show different behavioral profiles when they are exposed to flavor stimuli via maternal diet during pregnancy. The most used measures of infants' responses to prenatally exposed flavors were oro-facial responses ($k = 5$) using action units based on muscular activation [42]. Infant behavioral responses in the studies reviewed were mainly recorded offline and later analyzed by trained coders who were blind to the hypotheses, condition, and type of stimulus. Only one study [32] did not provide information about whether the behaviors were coded offline.

*Flavors* of alcohol ($k = 2$), anise ($k = 1$), carrot ($k = 1$), garlic ($k = 1$), green vegetables ($k = 1$) through maternal ingestion of foods or beverages during pregnancy were analyzed. The two

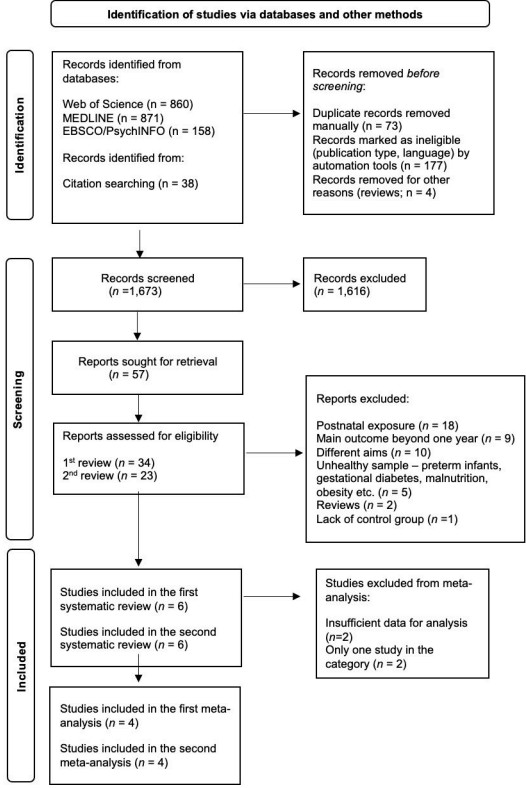

**Fig 1. Flow diagram of studies.**

studies on alcohol flavor reported that all infants were healthy during the pre and postnatal periods. Despite showing evidence of the transfer of alcohol flavor from mother to fetus during pregnancy, two studies [30, 31] were excluded from the meta-analysis due to insufficient data [30] to compute an effect size and for being the only study that measured appetitive responses [31]. These two studies presented a series of 11 odor stimuli to test the odor-elicited reactions. The study [31] reported that babies born to frequent drinkers and stimulated with ethanol on 10 of 11 trials (EtOH–Lem–EtOH sequence) exhibited significantly higher frequencies of appetitive responses relative to babies born to infrequent drinkers ($p < .025$), and our analysis produced the following individual effect size, $d = 1.368$ (0.476). Regarding the *timing* of prenatal flavor exposure, fetuses were exposed to specific target flavors mostly in the last two months of pregnancy. Only two studies [30, 31] measured the weekly flavor exposure throughout pregnancy. In terms of the *amount*, two studies [33, 34] determined a minimum amount of flavor exposure four days a week. The *duration* of the flavor exposure in the studies was between one week [32] and one month [33]. Examining studies, in terms of the *measurement of flavor consumption*, four observational studies measure the consumption of target flavor in the mothers' diet [30–32, 35]. One study [33] gave the target flavor directly to mothers to eat. One study [34] grouped the mothers based on their habitual intake of target flavor. In this study, the mothers were assigned to the experimental group if their diet involves the consumption of the target flavor. Experimental group mothers consumed the target flavor experimentally during the study. Total intake was calculated based on this experimental consumption as well as self-reported food consumption including target flavor.

**Table 2. Studies that compared the behavioral responses of infants whose mothers did or did not ingest the target flavor (*k* = 6).**

| Reference (First author, year, country) | Sample size | Gestational age during exposure | Flavor ingested by mothers | Stimuli used at infant testing | Infant age at testing | Outcome | Infant behavioral measurement | Effect size Cohen's d (SEd) | Covariance and confounders controlled |
|---|---|---|---|---|---|---|---|---|---|
| Faas 2000 [30] (Argentina) | 50 mother-infant dyads (17 exposed, 33 non-exposed) | During pregnancy | Alcohol flavour | Ethanol odour | 24–48 h | When the primary stimulus was ethanol, new-borns whose mothers were frequent drinkers had significantly higher head and facial movements in response to ethanol (p < .05). | Head orientation and facial responses Not included in the meta-analysis (insufficient data to compute an effect size) A set of 11 odor stimuli were given, primarily ethanol (EtOH-Lem-EtOH) or primarily lemon (Lem-EtOH-Lem). The first and last odours were provided five times in a sequence, but the middle (dishabituation) odour was given only once. | | |
| Faas 2015 [31] (Argentina) | 43 mother-infant dyads (16 exposed, 33 non-exposed) | During pregnancy | Alcohol flavour | Ethanol odour | 7–14 d | When the primary stimulus was ethanol, new-borns of frequent drinkers showed significantly higher frequencies of appetitive reactions to the ethanol odour in comparison to new-borns of infrequent drinkers (*p* < .03). Duration of appetitive responses, frequency of aversive responses towards ethanol sequence were not significantly affected by maternal alcohol intake. | Appetitive responses (frequency) Not included in the meta-analysis, insufficient number of studies in the category. These measurements were not included in the meta-analysis (insufficient data to compute an effect size). | 1.368 (0.476) | Gestational age at birth, infant sex, birthweight, birth height, delivery type, head circumference, Apgar scores, maternal age, parity, maternal age, infant age at assessment. |
| Hepper 1995 [32] (Northern Ireland) | 20 mother-infant dyads (10 exposed, 10 non-exposed) | In the last month of pregnancy | Garlic flavor | Garlic odor | 15–28 h | New-born exposed to garlic flavor oriented their head towards the garlic odor for longer (p = .016). | Head orientation (duration) | 1.189 (0.485) | Infant age at testing. |
| Mennella 2001 [33] (USA)* | 29 mother-infant dyads (15 exposed, 14 non-exposed) | Three consecutive weeks during last trimester of pregnancy | Carrot juice | Carrot flavor | 5.7 mo | Infants exposed to carrot juice had fewer negative facial expressions to carrot-flavored cereal (p < .05). | Negative facial responses (frequency) | -0.152 (0.392) | Race, singletons vs twins, breastfeeding, maternal age, infant age at testing, infant BMI, infant sex, mothers' eating habits. |

*(Continued)*

**Table 2.** (Continued)

| Reference (First author, year, country) | Sample size | Gestational age during exposure | Flavor ingested by mothers | Stimuli used at infant testing | Infant age at testing | Outcome | Infant behavioral measurement | Effect size Cohen's d (SEd) | Covariance and confounders controlled |
|---|---|---|---|---|---|---|---|---|---|
| Schaal 2000 [34] (France) | 23 mother-infant dyads (Day1:11 exposed, 12 non-exposed; Day4: 10 exposed, 10 non-exposed) | In the last two gestational weeks | Anise flavor | Anise odor | 8 h–4 d | New-born (day1) exposed to anise flavor had fewer negative facial responses in response to anise odor ($p < .001$). | Negative facial responses (frequency/day1) | -1.717 (0.368) | Gestational age at birth, delivery type, parity, maternal age, Apgar scores, infant sex, birthweight. |
| | | | | | | New-born (day4) exposed to anise in utero oriented their head towards the anise odor for longer ($p < .001$). | Head orientation (duration/day4) | 1.287 (0.491) | |
| | | | | | | New-born (day1) exposed to anise flavor had longer mouthing responses in response to anise odor ($p = .02$). | Mouthing (duration /day1) | 1.084 (0.458) | |
| Wagner 2019 [35] (France) | 79 mothers-infant dyads | In the last two months | Green vegetables | 2-isobutyl-3-methoxypyrazine | 8–12 mo | At 8 months, neonates whose mothers consumed more green vegetables in pregnancy showed higher liking scores for the corresponding odor** ($p < .001$). | Mouthing (duration) | 0.629 (0.240) | Maternal age, infant age, oronasal affections, infant gender, feeding style, breastfeeding |

Note.

* The analysis on the lactation period were not included in the review due to our aims.

** 2-isobutyl-3-methoxypyrazine, trimethylamine odorant corresponds to green vegetable food category.

In the meta-analysis ($k = 4$), there were 151 infants aged from 8 hours to 8 months showing reactions to specific flavors in three subcategories: reactions via the frequency of negative facial expressions, via the duration of head orientation and via the duration of mouthing behavior towards the odor exposed during pregnancy.

**The effects of prenatal flavor exposure on frequency of infant negative facial expressions.** Facial configurations of nose wrinkling, brow lowering, upper lip raising, lip corner-depressing, lip stretching, gaping and head-turning away from a stimulus are defined as negative facial responses in newborns, and are considered to express aversion towards a stimulus [43, 44]. We hypothesized that infants who had been exposed to specific flavor in utero would postnatally show fewer negative facial responses to the matching flavor, compared to infants whose mothers did not ingest the flavor during pregnancy.

Two studies [33, 34] were included in the meta-analysis measuring the frequency of negative facial expressions to the flavors that they were exposed to in the last trimester of pregnancy. Across these two studies, 52 infants (26 exposed, 26 not exposed) between 0.5 hours and 5.7 months old were assessed on the frequency of negative facial responses towards the flavor and odor stimuli that they experienced prenatally. Individual study effect sizes were -0.15 (CI = -0.921 to 0.616, $p = .70$) [33] and -1.72 (CI = -2.906 to -0.527, $p \leq .005$) [34].

**Table 3. Studies that investigated whether infants have different behavioral profiles to the odor of their own amniotic fluid (AF) compared to unfamiliar AF or control** *(k = 6)*.

| Reference (First, author, year, country) | Sample size | Attraction response to | Infant age at testing | Compared with (Control condition) | Outcomes | Infant behavioral measurement | Effect size Cohen's d (SEd) | Covariance and confounders controlled |
|---|---|---|---|---|---|---|---|---|
| Marlier 1998a [36] (France) | 38 mother-infant dyads | The odor of own AF | 2–4 d | Distilled water | Infants head orientation was significantly longer duration towards own AF than distilled water (*p* < .001). | Head orientation (duration) | 1.067 (0.203) | Gestational age at birth, infant sex, ethnicity, Apgar scores, birthweight, delivery type, maternal age, maternal socioeconomic level, feeding type, parity, arousal level of infants at testing. |
| Marlier 1998b [37] (France) | 22 mother-infant dyads | The odor of own AF | 15–57 h | Distilled water | Infants head orientation was significantly longer towards own AF than distilled water (*p* < .01). | Head orientation (duration) | 0.902 (0.253) | Gestational age at birth, infant sex, delivery type, Apgar scores, birthweight, maternal age, parity, socioeconomic level, breastfeeding, testing time, infants' hunger/satiety and arousal level. |
| Schaal 1995 [38] (France) | 37 mother-infant dyads | The odor of own AF | 13–47 h | Distilled water | Infants head orientation was significantly longer towards own AF than distilled water (*p* < .0007) | Head orientation (duration) | 0.649 (0.181) | Gestational age at birth, infant sex, delivery type, Apgar scores, birthweight, maternal age, parity, ethnicity, smoking status, socioeconomic level, breastfeeding, feeding type, age at testing. |
| Schaal 1998 [39] (France) | 12 mother-infant dyads | The odor of own AF | 48–96 h | Non-familiar AF | New-born turned their nose significantly longer to their own AF than non-familiar AF (*p* = .01). | Head orientation (duration) | 0.882 (0.34) | Gestational age at birth, infant sex, delivery type, Apgar scores, birthweight, maternal age, parity, infant age at test, breastfeeding, time on testing day, hunger/satiety and arousal level. |
| Varendi 1996 [40] (France) | 30 mother-infant dyads | The odor of breast moistened with their own AF | 6–23 min | Breast not moistened with own AF | A significant majority of new-born chose to feed on the breast moistened with their own AF (p < .001). | Head orientation (duration) | Not included in the meta-analysis (insufficient data to compute an effect size) | |
| Varendi 1998 [41] (France) | 32 mother-infant dyads | The odor of own AF | 31–90 min | No odor | The crying times were significantly shorter in AF odor group than control group (*p* = .02). | Crying (duration) | 0.891 (0.371) | Maternal smoking status delivery type, infant axillary temperature, parity, maternal age, infant sex, birthweight, infant age at test. |

*Note.* AF: Amniotic fluid.

Because of the heterogeneity in the effect sizes between the studies (Q = 4.69, *p* = .030, $I^2$ = 78.68%), the random effects model is reported. The combined effect size for negative facial responses is not significant (*d* = -0.87, 95% CI = −2.39 to 0.66; Z = -1.11, *p* = .266). This result shows that the frequency of negative facial responses towards the target flavor was not significantly different between infants prenatally exposed to those flavors and infants who were not exposed (see Fig 2).

**The effects of prenatal flavor exposure on duration of infant head orientation.** Two studies [32, 34] were included in the meta-analysis of the duration of head orientation towards the odor that was exposed during pregnancy. Across the two studies, 40 infants (20 exposed, 20 not exposed) between 15 and 110 hours old were assessed, by using a two-odor choice test, on the duration of head orientation towards different odors. Individual study effect sizes were

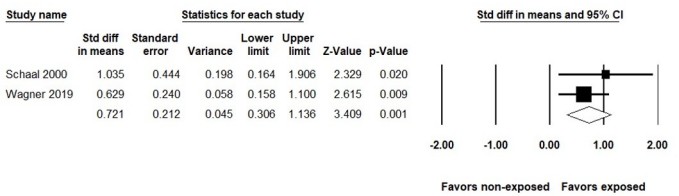

**Fig 2. Forest plot of meta-analysis in the first group of studies.** Diamonds represent the overall effect sizes, with squares representing individual studies. The size of the squares represents the weights assigned to each study.

1.19 (CI = 0.238 to 2.14, $p$ = 0.014) [32] and 1.29 (CI = 0.324 to 2.25, $p$ = 0.009) [34]. There is no evidence of heterogeneity (Q = 0.02, $p$ = .887, $I^2$ < 0.01%), and therefore, the fixed effect model is reported. The combined effect size for head orientation is significant ($d$ = 1.24; 95% CI = 0.56 to 1.91; Z = 3.58, $p$ < .001). This demonstrates that infants prenatally exposed to odors through maternal diet had significantly longer duration of head orientation towards the same olfactory stimulus (see Fig 2).

**The effects of prenatal flavor exposure on duration of infant mouthing behavior.** Two studies [34, 35] were included in the meta-analysis of the duration of mouthing behavior to the prenatally exposed flavors. A total of 102 infants between 8 hours and 8 months old were assessed on the effects of anise or green vegetable exposure. Individual study effect sizes were 0.629 (CI = 0.158 to 1.1, $p$ = .009) [35] and 1.035 (CI = 0.164 to 1.906, $p$ = .02) [34]. There is no evidence of heterogeneity (Q = 0.646, $p$ = .421, $I^2$ < 0.01%), and therefore, the fixed effect model is reported. The pooled effect size for mouthing behavior is significant ($d$ = 0.72; 95% CI = 0.306 to 1.136; Z = 3.409, $p$ ≤ .001) demonstrating longer duration of mouthing behaviors to prenatally exposed flavors (see Fig 2).

## Association between the odor of amniotic fluid and infant behavioral profile

There are five studies that have investigated whether infants have different behavioral profiles to the odor of their own amniotic fluid that they have been exposed to throughout gestation

(see Table 3). Five observational studies [36–40] assessed infant reactions to the familiar amniotic fluid measuring the duration of the head orientation towards the familiar stimulus, and one randomized control study [41] measured the duration of crying. To measure head orientation towards a familiar amniotic fluid odor, all studies used a two-choice test involving the presentation of two stimuli placed symmetrically on either side of the infant's head. As a control stimulus, studies used distilled water ($k = 3$), non-familiar amniotic fluid ($k = 1$), the natural odor of the mother's breast ($k = 1$), and no odor stimulus ($k = 1$).

All studies except one [40] measured only nasal chemoreception. In that study, infants were able to lick (oral response) and/or smell (nasal response) the mother's breast which was moistened by their own amniotic fluid. However, this study was not included due to insufficient data to compute the effect size, although they reported that a majority of new-born chose to feed on the breast moistened with their own amniotic fluid. Also, the study that measured the duration of crying [41] was not included due to being the only study with this behavioral outcome. This study did however find a significant effect—crying times were significantly shorter in the amniotic fluid odor group than in the control group ($d = - 0.891$, $p = .02$).

In the meta-analysis, a total of 109 infants between five and 96 hours old were assessed by a two-choice odor test, on the duration of head orientation towards their own amniotic fluid odor paired with a control condition (e.g., distilled water, unfamiliar amniotic fluid or natural mother's breast odor). Individual study effect sizes ranged between 0.649 [38] and 1.067 [36]. The effect sizes in these studies are homogeneous ($Q = 2.417$, $p = .49$, $I^2 < 0.01\%$), and therefore, the fixed effect model is reported. The combined effect size for the duration of head orientation is significant ($d = 0.853$; 95% CI = .632 to 1.073; $Z = 7.58$, $p < .001$, fail-safe N = 55). Visual inspection of the funnel plot (see S1 Fig) suggested a relatively symmetric distribution of study findings. Results showed that infants oriented their heads for significantly longer towards their own amniotic fluid compared to the control condition during the first two postnatal days (see Fig 3).

## Discussion

The aim of this systematic review and meta-analysis was to examine whether there is a transnatal chemosensory continuum from fetal to the first year after birth. To answer this question, infant responses to flavors transferred via maternal diet prenatally and infant responses to their own amniotic fluid odor were analyzed separately to distinguish between different (but related) media of transmission. Overall, the results presented here indicate that there is a chemosensory continuity from prenatal to the first year of postnatal life, although the effect of flavor exposure through maternal diet was not consistent across all types of postnatal reactions.

Studies included in the first group highlight that the effects of maternal consumption of specific flavors, including alcohol, carrot, anise, garlic and green vegetables during pregnancy influence infant responses to these flavors hours, days, and months after birth. However, the results from the current work cannot be generalized to different types of flavours because food flavour molecules are metabolised in unique patterns due to their diverse chemical compositions [45]. Although, we cannot extrapolate from this finding to all flavors that the mother consumes during pregnancy, it supports the current evidence [3, 8] that maternal dietary aromas are transferred to the fetal environment and that these flavor cues are sensed by fetuses and later accepted by infants. Thus, manipulating maternal diet can shape postnatal food preferences [46, 47]. All the studies included in this review and meta-analysis recruited samples of healthy infants from healthy pregnant mothers. The behavioral responses of the infants to the transferred flavours from the maternal diet can therefore only be generalised to a healthy infant population.

## Association between the odor of amniotic fluid and duration of head orientation

| Study name | | | | | | | | Std diff in means and 95% CI |
|---|---|---|---|---|---|---|---|---|
| | Std diff in means | Standard error | Variance | Lower limit | Upper limit | Z-Value | p-Value | |
| Marlier 1998a | 1.067 | 0.203 | 0.041 | 0.668 | 1.465 | 5.249 | 0.000 | |
| Marlier 1998b | 0.902 | 0.253 | 0.064 | 0.406 | 1.397 | 3.567 | 0.000 | |
| Schaal 1995 | 0.649 | 0.181 | 0.033 | 0.295 | 1.004 | 3.589 | 0.000 | |
| Schaal 1998 | 0.882 | 0.340 | 0.116 | 0.215 | 1.549 | 2.592 | 0.010 | |
| | 0.853 | 0.112 | 0.013 | 0.632 | 1.073 | 7.580 | 0.000 | |

Favors control    Favors amniotic fluid

**Fig 3. Forest plot of meta-analysis in the second group of studies.** Diamonds represent the overall effect sizes, with squares representing individual studies. The size of the squares represent the weights assigned to each study.

In the meta-analysis of postnatal reactions to prenatally introduced flavors via maternal consumption of flavors, there were three subcategories: frequency of negative facial expressions, duration of head orientation, and duration of mouthing behavior and. The studies compared infants who were exposed to a specific flavor at any point during pregnancy to infants whose mothers did not consume that flavor. Most studies did not use randomized designs. Prenatal flavor exposure through maternal diet resulted in a large significant effect in longer duration of head orientation [32, 34] and mouthing behavior [34, 35]. to the familiar flavor, which can be considered to represent preference, attraction, or acceptance [19–22].

In contrast, the combined effect size of the other two studies that looked at negative facial expressions was non-significant. One of the two studies [34] in that category found a significant individual effect size indicating that those infants born to mothers who ingested anise flavor in the last two weeks of pregnancy had fewer negative facial expressions after birth to anise odor. Although infants whose mothers consumed carrot juice showed fewer negative facial expressions towards carrot cereal over to plain cereal at ~6 month [33], the effect size of the comparison between exposed and unexposed infants, the comparators included in our meta-analysis, when fed with carrot-flavored cereal [33] was not significant. This result might be explained with the difference between the type of stimuli used at infant testing. Schaal et al. [34] measured infant olfactory responses whereas Mennella et al. [33] assessed gustatory responses. Furthermore, the potential difference in terms of discrete oro-facial movements added to the negative facial expression definition could be another explanation. Schaal et al. [34] analyzed brow lowerer, nose wrinkle, upper lip raiser, lip corner depressor, lip stretch, mouth stretch, jaw drop and head turning, whereas Mennella et al. [33] only provided examples of movements (brow lowerer, nose wrinkle, upper lip raiser, mouth stretch, jaw drop and head turning) and were included into the negative facial configurations. This highlights us the importance of using an objective coding scheme while analyzing human reactions to stimuli in order to be consistent across different studies. Despite the nonsignificant results between groups, we could not dismiss the individual effect size in one of the studies [34] as well as the significant effect reported that infants exposed to carrot in the amniotic fluid preferred carrot-flavored cereal over plain cereal [33]. Previous research indicated that negative facial configurations are more discriminating than positive ones, particularly when evaluating neonatal

odor-elicited behaviors [19–20, 44, 48]. Hence, we may still argue that negative facial expressions are an effective method for measuring newborns' hedonic reactions.

The second group of studies increased our further understanding of the chemosensory continuum from prenatal to postnatal life by analyzing infant reactions to the odor of their own amniotic fluid, this has not been examined in previous systematic reviews. This analysis showed that when infants are introduced postnatally to their own amniotic fluid odor, they displayed longer head orientation and shorter crying periods from birth to four postnatal days.

Corroborating previous research findings [1, 7, 49], these results showed that fetal chemosensory abilities are functional to detect olfactory molecules in their own amniotic fluid [1, 11, 20] and that these prenatal olfactory molecules were recalled after birth, resulting in soothing effect and preference for familiar chemosensory inputs. Furthermore, fetal nasal perception was sufficient to mediate the neonatal selective response because all studies included in the meta-analysis measured responses to olfactory stimuli.

The results indicate that human fetuses can detect flavor signals from their mothers' diets as well as the odor of their own amniotic fluid and embed information during pregnancy for postnatal use. Repeated exposure to specific flavor through maternal consumption during the last two months of pregnancy is the optimal period to have the effects on postnatal behaviors. The effects of prenatal chemosensory inputs on postnatal outcomes were observed mostly in the first couple of weeks after birth but there were also effects after 6 and 8 months of life.

## Strengths and limitations

This review advances current knowledge of the impact of the prenatal environment on postnatal life. The current systematic review is the first study which uses meta-analysis to examine whether there is transnatal chemosensory continuity from fetal to neonatal life by focusing on two related fields of research. We applied a six-component search strategy spanning four databases, as well as manual searches, to identify all relevant academic articles on this topic. The review does, however, have limitations, including the quantity of studies and confounding factors. Firstly, only four different studies were included in each group of meta-analysis because of a lack of studies in the subcategory or insufficient data reported. Secondly, individual variables such as fetal sex, birth outcomes, genetic determinants of chemosensory perception, maternal eating habits, socio-economic status, and ethnicity may affect the individual response to prenatal chemosensory inputs [50–56]. Additionally, because we included all studies investigating infant reactions up to one year old, we must consider the probability that flavor exposure during breastfeeding and/or complementary feeding can influence infant responses [17, 22, 57–59]. However, in all included studies, infant testing was completed when they were under the age of six months, which minimizes the effects of complementary feeding considering that most infants have probably not yet been introduced to solid foods at this age [58–60].

## Research recommendations

Measuring postnatal behavioral reactions to prenatally exposed flavors, odors, or tastes may provide indirect evidence of flavor transfer from maternal diet to fetal environment, fetal chemosensory abilities, fetal memory for flavors and thus chemosensory continuity from prenatal to postnatal life. Future studies investigating fetal behavior directly (e.g., via 4D ultrasound scanning) are required to understand how a fetus perceives and responds to the prenatal flavor environment.

By providing the evidence of chemosensory continuity from fetal to neonatal life, it can be argued that exposing pregnant women to diverse and healthy vegetables is a potentially plausible way to improve lifelong health and drive healthy choices in populations widely concerned

by the obesity pandemic by facilitating the process of infant vegetable acceptance. Furthermore, the findings provide evidence that repeated flavour exposure facilitates the plasticity of preferences acquired in utero, and thus the results have important implications for our understanding of perinatal continuity in stimuli perception and memory from fetal to neonatal life. Longitudinal prospective studies and randomised clinical trials starting from fetal stage until infancy, childhood and adulthood are required to explore chemosensory continuity over the life span.

The current study allows us to argue that fetuses are not protected from maternal food choices, a situation exposing them inescapably to the environmental regimen mediated by the mother's body. Thus, this study call to research on materno- fetal flavour transfer of other ingested or inhaled compounds and their possible long-term effects relating to the concept of food-related behaviours.

*In conclusion*, the results of this systematic review and meta-analysis support the hypothesis that there is transnatal continuity between prenatal and postnatal life.

## Supporting information

**S1 Checklist. Prisma checklist.**
(DOC)

**S1 Table. Quality assessment of included studies.**
(PDF)

**S1 Fig. Funnel plot for the association between the odor of amniotic fluid and the duration of head orientation.**
(TIF)

## Acknowledgments

The review was not registered.

## Author Contributions

**Conceptualization:** Beyza Ustun, Judith Covey.

**Data curation:** Beyza Ustun.

**Formal analysis:** Beyza Ustun.

**Funding acquisition:** Beyza Ustun.

**Investigation:** Beyza Ustun.

**Methodology:** Beyza Ustun, Judith Covey.

**Project administration:** Beyza Ustun.

**Resources:** Beyza Ustun.

**Software:** Beyza Ustun.

**Supervision:** Judith Covey.

**Validation:** Beyza Ustun, Judith Covey.

**Visualization:** Beyza Ustun.

**Writing – original draft:** Beyza Ustun.

**Writing – review & editing:** Judith Covey.

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
