## [Decision Letter · Decision Letter 0]

16 Jan 2023

PONE-D-22-32981Chemosensory continuity from prenatal to postnatal life in humans: A systematic review and meta-analysisPLOS ONE

Dear Dr. Ustun,

Thank you for submitting your manuscript to PLOS ONE. After careful consideration, we feel that it has merit but does not fully meet PLOS ONE’s publication criteria as it currently stands. Therefore, we invite you to submit a revised version of the manuscript that addresses the points raised during the review process.

We look forward to receiving your revised manuscript.

Kind regards,

Sawsan Abuhammad

Academic Editor

PLOS ONE

Journal Requirements:

"This study was undertaken as part of a Ph.D. thesis funded by the Turkish Ministry of 

National Education. The funder has not had any role in the study protocol, analysis, or 

preparation of the manuscript. The review was not registered."

"This study was undertaken as part of a Ph.D. thesis funded by the Turkish Ministry of National Education. The funder has not had any role in the study protocol, analysis, or preparation of the manuscript."

"The author(s) declared that there were no conflicts of interest with respect to the

authorship or the publication of this article."

Reviewers' comments:

Reviewer's Responses to Questions

**Comments to the Author**

1. Is the manuscript technically sound, and do the data support the conclusions?

Reviewer #1: Yes

2. Has the statistical analysis been performed appropriately and rigorously? 

Reviewer #1: Yes

3. Have the authors made all data underlying the findings in their manuscript fully available?

Reviewer #1: Yes

4. Is the manuscript presented in an intelligible fashion and written in standard English?

Reviewer #1: Yes

5. Review Comments to the Author

Reviewer #1: Applicability of results:

Even if the results of a meta-analysis are statistically significant, they must be useful in clinical practice or serve as a message for researchers planning future investigations. The findings must have external validity or generalizability and must have an influence on the management of a specific patient. Furthermore, the research included in the metaanalysis should contain infants groups observed in actual practice.

Clinical relevance of results:

The final stage in thoroughly evaluating a meta-analysis should be analyzing the clinical usefulness of the data. The results of a meta-analysis may be statistically significant yet have a little practical impact.

6. PLOS authors have the option to publish the peer review history of their article (what does this mean?). If published, this will include your full peer review and any attached files.

Reviewer #1: **Yes: **Alaa Dalky

---

## [Author Response · Author response to Decision Letter 0]

27 Feb 2023

We would like to thank you for the opportunity to revise and resubmit our manuscript. The comments from you and the reviewer were helpful in revising the manuscript and we have carefully considered and responded to each suggestion in the rebuttal letter. Corresponding changes are highlighted in the marked-up copy of our manuscript with track changes. 

Journal Requirements:

Comment 1. When submitting your revision, we need you to address these additional requirements.

Please ensure that your manuscript meets PLOS ONE's style requirements, including those for file naming. The PLOS ONE style templates can be found at https://journals.plos.org/plosone/s/file?id=wjVg/PLOSOne_formatting_sample_main_body.pdf and https://journals.plos.org/plosone/s/file?id=ba62/PLOSOne_formatting_sample_title_authors_affiliations.pdf .

Response 1: The manuscript has been revised according to the journal style requirements (please see the revised manuscript).

Comment 2. Thank you for stating the following in the Acknowledgments Section of your manuscript: 

"This study was undertaken as part of a Ph.D. thesis funded by the Turkish Ministry of National Education. The funder has not had any role in the study protocol, analysis, or preparation of the manuscript. The review was not registered."

"This study was undertaken as part of a Ph.D. thesis funded by the Turkish Ministry of National Education. The funder has not had any role in the study protocol, analysis, or preparation of the manuscript."

Response 2: The funding statement has been removed from the revised manuscript. We would like to keep the current funding statement as follows: "This study was undertaken as part of a Ph.D. thesis funded by the Turkish Ministry of National Education. The funder has not had any role in the study protocol, analysis, or preparation of the manuscript."

Comment 3. Thank you for stating the following in your Competing Interests section: "The author(s) declared that there were no conflicts of interest with respect to the authorship or the publication of this article."

Response 3: The declaration of competing interest has been removed from the revised manuscript. We have stated “The authors have declared that no competing interests exist” in our cover letter. 

Comment 4. We note that you have stated that you will provide repository information for your data at acceptance. Should your manuscript be accepted for publication, we will hold it until you provide the relevant accession numbers or DOIs necessary to access your data. If you wish to make changes to your Data Availability statement, please describe these changes in your cover letter and we will update your Data Availability statement to reflect the information you provide. 

Response 4: All relevant data are within the manuscript and its Supporting Information files.

Comment 5. Please include captions for your Supporting Information files at the end of your manuscript, and update any in-text citations to match accordingly. Please see our Supporting Information guidelines for more information: http://journals.plos.org/plosone/s/supporting-information.

Response 5: The manuscript has been revised according to the supporting information guidelines (please see the revised manuscript).

Comment 6. Please review your reference list to ensure that it is complete and correct. If you have cited papers that have been retracted, please include the rationale for doing so in the manuscript text, or remove these references and replace them with relevant current references. Any changes to the reference list should be mentioned in the rebuttal letter that accompanies your revised manuscript. If you need to cite a retracted article, indicate the article’s retracted status in the References list and also include a citation and full reference for the retraction notice.

Response 6: The reference list has been reviewed and we ensure that it is complete and correct. 

Response to Reviewer #1 comments:

Comment 1: Applicability of results:

Even if the results of a meta-analysis are statistically significant, they must be useful in clinical practice or serve as a message for researchers planning future investigations. The findings must have external validity or generalizability and must have an influence on the management of a specific patient. Furthermore, the research included in the metaanalysis should contain infants groups observed in actual practice.

Response 1: We would like to thank the reviewer for the useful remarks. Regarding the generalizability, the following sentence has been added to the manuscript: 

“All the studies included in this review and meta-analysis recruited samples of healthy infants from healthy pregnant mothers. The behavioral responses of the infants to the transferred flavours from the maternal diet can therefore only be generalised to a healthy infant population” (page 23, line 332-334). 

Additionally, as we stated in the discussion (page 22, line 323-325), we found that maternal consumption of five specific flavours during pregnancy influence infant behavioural outcomes, but it is difficult to generalise this result to all flavours because different flavours might have different transfer mechanisms in the prenatal environment. Thus, the following sentence has been added to the manuscript: 

“However, the results from the current work cannot be generalized to different types of flavours because food flavour molecules are metabolised in unique patterns due to their diverse chemical compositions [45].” (page 22, line 325-328). 

Comment 2: Clinical relevance of results:

The final stage in thoroughly evaluating a meta-analysis should be analyzing the clinical usefulness of the data. The results of a meta-analysis may be statistically significant yet have a little practical impact. 

Response 2: The corresponding excerpts from the revised manuscript follow (changes highlighted in yellow): (page 26 line 408 – page 27 line 422). 

“By providing the evidence of chemosensory continuity from fetal to neonatal life, it can be argued that exposing pregnant women to diverse and healthy vegetables is a potentially plausible way to improve lifelong health and drive healthy choices in populations widely concerned by the obesity pandemic by facilitating the process of infant vegetable acceptance. Furthermore, the findings provide evidence that repeated flavour exposure facilitates the plasticity of preferences acquired in utero, and thus the results have important implications for our understanding of perinatal continuity in stimuli perception and memory from fetal to neonatal life. Longitudinal prospective studies and randomised clinical trials starting from fetal stage until infancy, childhood and adulthood are required to explore chemosensory continuity over the life span. 

The current study allows us to argue that fetuses are not protected from maternal food choices, a situation exposing them inescapably to the environmental regimen mediated by the mother’s body. Thus, this study call to research on materno- fetal flavour transfer of other ingested or inhaled compounds and their possible long-term effects relating to the concept of food-related behaviours.”

Additional revisions from the authors to help orient the reader: highlighted in blue in the revised manuscript.

1. The following sentences have minor changes for clarification: 

“This review advances current knowledge of the impact of the prenatal environment on postnatal life.” (page 25, line 385-386). 

“Future studies investigating fetal behavior directly (e.g., via 4D ultrasound scanning) are required to understand how a fetus perceives and responds to the prenatal flavor environment.” (page 26, line 405-406).

2. We have added the ref number 45 (Jeleń, 2012) in the revised manuscript to make our justification clear in the text.

---

## [Editor Report · Decision Letter 1]

6 Mar 2023

Chemosensory continuity from prenatal to postnatal life in humans: A systematic review and meta-analysis

PONE-D-22-32981R1

Dear Dr. Ustun,

We’re pleased to inform you that your manuscript has been judged scientifically suitable for publication and will be formally accepted for publication once it meets all outstanding technical requirements.

Kind regards,

Sawsan Abuhammad

Academic Editor

PLOS ONE
---

## [Editor Report · Acceptance letter]

20 Mar 2023

PONE-D-22-32981R1 

Chemosensory continuity from prenatal to postnatal life in humans: A systematic review and meta-analysis 

Dear Dr. Ustun:

I'm pleased to inform you that your manuscript has been deemed suitable for publication in PLOS ONE. Congratulations! Your manuscript is now with our production department. 

Kind regards, 

on behalf of

Dr. Sawsan Abuhammad 

Academic Editor

PLOS ONE